# Temporal Spiking Generative Adversarial Networks for Heading Direction Decoding

## Abstract

The spike-based neuronal responses within the ventral intraparietal area (VIP) exhibit intricate spatial and temporal dynamics in the posterior parietal cortex, presenting decoding challenges such as limited data availability at the biological population level. The practical difficulty in collecting VIP neuronal response data hinders the application of sophisticated decoding models. To address this challenge, we propose a unified spike-based decoding framework leveraging spiking neural networks (SNNs) for both generative and decoding purposes, for their energy efficiency and suitability for neural decoding tasks. We propose the Temporal Spiking Generative Adversarial Networks (T-SGAN), a model based on a spiking transformer, to generate synthetic time-series data reflecting the neuronal response of VIP neurons. T-SGAN incorporates temporal segmentation to reduce the temporal dimension length, while spatial self-attention facilitates the extraction of associated information among VIP neurons. This is followed by recurrent SNNs decoder equipped with an attention mechanism, designed to capture the intricate spatial and temporal dynamics for heading direction decoding. Experimental evaluations conducted on biological datasets from monkeys showcase the effectiveness of the proposed framework. Results indicate that T-SGAN successfully generates realistic synthetic data, leading to a significant improvement of up to 1.75% in decoding accuracy for recurrent SNNs. Furthermore, the SNN-based decoding framework capitalizes on the low power consumption advantages, offering substantial benefits for neuronal response decoding applications.

## 1 Introduction

The neuronal responses of the ventral intraparietal area (VIP) suggest pronounced spatial and temporal dynamics (Bremmer et al., 2017; Chen et al., 2013). Decoding heading directions from VIP population responses with spike trains is crucial for understanding primates self-motion. Spiking neural networks (SNNs) are biologically plausible models known for their effectiveness in handling sequence data with rich spatiotemporal dynamics (Gerstner et al., 2014; Kumarasinghe et al., 2021). SNNs' communication through spike trains promotes spatiotemporal dynamics and energy efficiency, making them suitable for neuromorphic chips and energy efficient (Merolla et al., 2014; Davies et al., 2018; Pei et al., 2019). The SNN-based heading decoding model holds promise for improving spike-based neural population response decoding with low power consumption, leveraging the abundant spatiotemporal characteristics and event-driven nature of SNNs (Roy et al., 2019). Training SNNs requires diverse and stable data. However, acquiring VIP neuronal responses in monkeys is expensive and time-consuming, impeding the collection of sufficient spike-based data for effective SNN model training. The training of SNNs typically requires the stable and diverse dataset. However, acquiring VIP neuronal responses in monkeys is expensive and time-consuming, posing challenges in collecting sufficient spike-based data for effective SNN model training.

To alleviate the data acquisition requirements in training deep neural network models, generative models have successfully provided synthetic data, enhancing various applications in computer vision and natural language processing (Jing et al., 2019; Tao et al., 2022; Nichol et al., 2022). Generative Adversarial Networks (GANs), for instance, consist of a generator and a discriminator. The generator produces data matching the dimensions of real data, while the discriminator evaluates the authenticity of the generated data. Through adversarial training, the generator and discriminator strive for equilibrium, resulting in generated data with more diversity than real data to improve

model training. The proliferation of generative models has significantly contributed to synthetic data for neural decoding, supporting the advancement of techniques for clinical decision support by prototyping automated research workflows and addressing privacy concerns (Yan et al., 2022). However, existing generative models for neural decoding primarily focus on synthetic continuous biological data (Luo et al., 2020). While generative models based on traditional artificial neural networks for continuous time-series data have been extensively explored, the generation of synthetic spike-based data using SNNs to enhance the neural decoding of population neuronal responses remains an unexplored domain. Hence, we aim to enhance spike-based data decoding by introducing a spike-based GAN model, focusing specifically on decoding neural signals from the VIP population of neurons in response to heading directions.

Moreover, due to the highly dynamic and noisy nature of VIP neural responses in spatial and temporal dimensions, denoising irrelevant information and adaptively capturing crucial information become imperative. To solve this problem, we attempts attention mechanisms in SNNs to enhance critical feature extraction and improve the robustness of neural decoding models (Jiang et al., 2021b). HHowever, the integration of SNNs with self-attention remains underexplored in current literature. Recent works, such as the spike-driven Transformer (Yao et al., 2021), have showcased the energy efficiency of SNNs when processing temporal sequences. However, these studies primarily focus on SNNs' capabilities in handling sequences, neglecting the joint consideration of spike-based data generation and classification.

Therefore, in this study, we propose a comprehensive spike-based decoding framework that integrates both data generation and decoding of neuronal responses using SNNs with attention mechanisms. Our primary objective is to enhance the decoding of spike-based data. For synthetic spike-based data augmentation, the spike-based GANs framework is considered for its realistic data and relatively moderate computation cost. We build the temporal Transformer-based GANs with spike computation. With the sparse spike-form Query, Key, and Value, its computation becomes more efficient for the reduction of Multiply-and-Accumulate (MAC) calculations after introducing sparse spike-based operation. Thus, the generated spike train data by T-SGAN is used for data augmentation. For the following heading direction decoding component, we employ the spatial and temporal attention mechanism in recurrent SNNs decoder to adaptively select the pivotal time period and sensitive neurons for each heading direction and thus strengthen direction decoding. The above unified spike-based decoding framework could take the biological plausibility and low energy consumption advantages of SNNs for solving the biological neuronal response decoding problem.

To summarize the main contributions of this article:

- The paper proposes a spike-based unified decoding framework, comprising the temporal spiking generative adversarial networks (T-SGAN) for data augmentation and recurrent SNNs with spatial and temporal attention for robust decoding. This framework is designed to effectively map VIP population responses to heading behavior.

- The paper introduces a temporal segmentation in T-SGAN to shorten the temporal dimension and incorporates spatial self-attention to extract correlated information. This design enhances the generation of synthetic spike data that is both long and sparse.

- By incorporating the spatial and temporal attention mechanism into the recurrent SNNs decoder, the ReSNN decoder can adaptively select the pivotal time period and sensitive neurons for each heading direction and thus strengthen heading direction decoding performance.

- The experiments are conducted on the biological datasets from monkeys to evaluate the decoding performance of the proposed framework. The ablation study is carried out to indicate the effectiveness of the temporal segmentation method in T-SGAN. The results show that the proposed model achieves competitive decoding accuracy of 95.2% on dataset1 and 93.15% on dataset2 compared with other decoding models.

The experimental dataset is collected from rhesus monkeys by presenting them with eight linear space azimuth directions of vestibular heading stimuli. We also visualize the generated synthetic data and real data and use t-SNE to map the multi-dimensional output sequence vectors into two dimensions to visually observe the similarity and diversity between them.

## 2    RELATED WORKS

**Generative models based on SNNs.**  Generative models utilizing Spiking Neural Networks (SNNs) have been a relatively underexplored area in existing literature. The Spiking-GAN, introduced by Kotariya et al. (Kotariya & Ganguly, 2022), employs time-to-first-spike coding to address static image generation. This model extends the refractory period for integrate-and-fire neurons to ensure single-spike firing for each neuron. In a different approach, Rosenfeld et al. (Rosenfeld et al., 2022) proposed the SpikeGAN, which incorporates a hybrid SNN-ANN architecture. This model is designed to train SNNs to match distributions of spiking signals rather than individual spiking signals, supporting Bayesian learning for the generator's weights. Kamata et al. (Kamata et al., 2022) contributed to the field with the development of a fully spiking variational autoencoder for high-quality image generation. This model constructs the latent space using an autoregressive SNN, ensuring that latent variables follow a Bernoulli distribution for variational learning. Despite these advancements, the exploration of pure SNN-based generative models specifically tailored for long time-series neural signals remains largely unexplored in current research.

**GANs for Time Series Data Generation.**  In the realm of GANs based on ANNs, there is an ongoing effort to advance the generation of high-quality, diverse, and private time series data. Yoon et al. (2019) (Yoon et al., 2019) introduced Time GAN, presenting a framework that combines conventional unsupervised GAN training methods with a more controllable supervised learning approach. To address the challenges posed by the increased dimensionality of generative modeling for long time series data streams, SigCWGAN (Ni et al. 2020) (Ni et al., 2020) offers a solution. Meanwhile, SynSigGAN (Hazra et al., 2020) (Hazra & Byun, 2020) focused on generating various types of continuous physiological/biomedical signal data. While prior attempts at time-series GANs predominantly relied on Recurrent Neural Network (RNN)-based architectures, recent studies suggest that theoretically, transformer-based GANs should outperform them. Jiang et al. (2021) (Jiang et al., 2021a) presented a pure transformer GAN model for synthetic image generation, drawing inspiration from the Vision Transformer model in discriminator design. Li et al. (2022) (Li et al., 2022) proposed TTS-GAN, where both the generator and discriminator are transformer encoders. They treat a time-series sequence as an image with a height of 1, introducing a novel perspective to time-series data representation in GANs.

**Decoding models based on SNNs.**  Neural decoding models based on Spiking Neural Networks (SNNs) have garnered increasing attention for their biological plausibility and energy efficiency, especially when coupled with neuromorphic hardware. The potential of these models for effective clinical applications in the future is particularly promising. Kumarasinghe et al. (2021) (Kumarasinghe et al., 2021) introduced the Brain-Inspired Spiking Neural Network (BI-SNN) model, designed for incremental learning, to predict muscle activity and upper-limb kinematics from electroencephalograph signals. Integration of the BI-SNN model with the NeuCube SNN architecture, alongside the eSPANNet model, revealed the BI-SNN's superior performance as a neural decoder for non-invasive brain-computer interfaces (BCIs). In an effort to leverage spatial and temporal dependencies within EEG signals, Kumar et al. proposed SNNs incorporating spatial convolutional, temporal convolutional, and recurrent layers (Kumar et al., 2022). This SNN architecture was implemented on the Loihi neuromorphic processor, showcasing the effectiveness of SNNs in EEG decoding. The computational advantages demonstrated in this study hold promise for future portable BCI systems. Consequently, SNN models emerge as a highly suitable choice for decoding neural signals, particularly those associated with spike-based population responses.

## 3    METHODS

The unified spike-based decoding framework, illustrated in Figure 1, comprises two key components: Temporal Spiking Generative Adversarial Networks (T-SGAN) for augmenting VIP population response spike data and Recurrent Spiking Neural Networks (ReSNNs) tailored for robust decoding of VIP neuronal responses into heading directions.

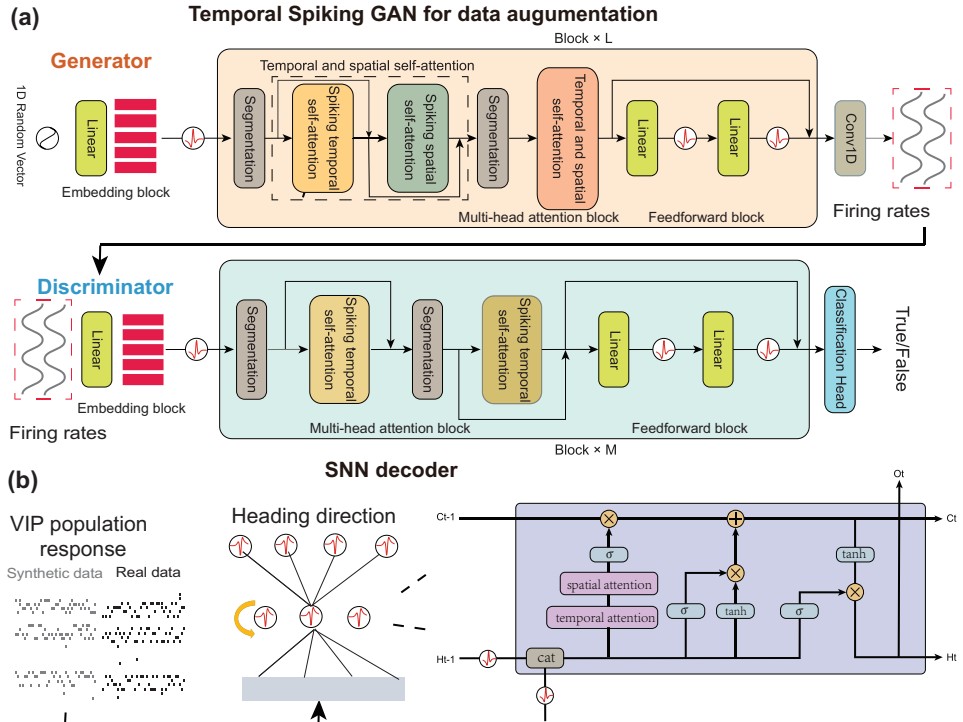

Figure 1: The architecture of the proposed unified spike-based decoding framework. The framework contains the T-SGAN for data augmentation of VIP population responses **(a)**, and the recurrent SNNs for robust decoding from VIP neuronal response to heading direction **(b)**. The T-SGAN consists of generator and discriminator based on the spike-based transformer structure which is composed of the embedding block, multi-head attention, and feedforward block. Meanwhile, the recurrent SNNs with spatial and temporal attention are employed to capture the corresponding features to improve the heading directions decoding performance.

## 3.1 THE T-SGAN MODEL

The T-SGAN model consists of a generator and a discriminator, both adopting a spike-based transformer architecture, illustrated in Figure 1 (a). This architecture includes three main components: an embedding block, multi-head attention, and a feedforward block. Normalization and dropout layers are strategically positioned before and after the multi-head attention and feedforward blocks, respectively. Residual connections have been incorporated within these two blocks.

Moreover, recognizing the significance of generated data in the training process for heading direction classification, we employ a spike-based Constrained GAN for the generation of synthetic data along with corresponding labels. This holistic T-SGAN model excels at efficient synthetic data generation, thereby significantly contributing to the improved decoding performance of the subsequent recurrent SNNs decoder, as depicted in Figure 1 (b).

### 3.1.1 DATA PROPROCESSING

Prior to inputting the original temporal sequence data into the GAN for training, we employ a sliding window approach by computing the firing rate of neurons within specific time bins. The sequences are evenly divided into multiple segments along the temporal dimension, each segment comprising $M$ time points corresponding to the chosen window size. To reduce the number of time points, vectors within the same segment are aggregated. Assuming we have an initial matrix with a shape of $(T, C)$, applying the window function yields a matrix with a shape of $(T//M, C)$, which is then fed into the GAN model during the training process. It is crucial to note that the window size $M$ should not be excessively large to ensure the capture of a more diverse set of temporal dynamics present in different sliding windows.

### 3.1.2 NETWORK ARCHITECTURES

**Generator.** The generator receives 1D vector $I \in R^D$. $I$ follows Gaussian distribution, $I_i \sim N(0, 1)$. $R$ and $D$ represent the batch size and the input latent dimension which is a configurable hyperparameter, respectively.

*Embedding block.* In the embedding block, the labels are initially input into the embedding layer. Subsequently, the embedded labels are concatenated with the input vector to generate a new vector. This new vector undergoes transformation into $R^{T \times E}$, where $T$ and $E$ represent the data length and embedding size, respectively. Trainable positional encoding values are then added. Following this, a leaky integrate-and-fire (LIF) neuron is employed to generate spiking data.

As the fundamental unit of SNNs, the postsynaptic spiking neuron in the LIF neuron model accumulates its membrane potential by receiving resultant currents from presynaptic neurons. Upon surpassing the firing threshold, the postsynaptic spiking neuron emits a spike. The computational formulation of the LIF neuron is described as follows:

$$H[t] = V[t-1] + \frac{1}{\tau}(X[t] - (V[t-1] - V_{reset})), \tag{1}$$

$$S[t] = \Theta(H[t] - V_{th}), \tag{2}$$

$$V[t] = H[t](1 - S[t]) + V_{reset}S[t], \tag{3}$$

where $X[t]$ is the input current at time step $t$ and $\Theta(v)$ is a firing function. When $v \geq 0$ the firing function outputs 1, otherwise outputs 0. $\tau$ is the membrane time constant and $V_{reset}$ represents resting membrane potential and $V_{th}$ is threshold potential. If the current potential is larger than $V_{th}$, the neuron fire a spike. Here for a simplification we set $V_{reset} = 0$ and $V_{th} = 1$.

Before each multi-head attention block, we reshape the transformed data to execute temporal dimension segmentation (Details about this will be seen in Section 3.1.3).

*Multi-head attention and feedforward blocks.* Inside each multi-head attention block, we first apply temporal self-attention component to extract temporal information. This component is a normal vanilla self-attention combining spiking mechanism and processes data within a segment which is obtained after temporal dimension segmentation. The spatial self-attention component follows by the temporal self-attention components after the segmentation component (See details in Section 3.1.4). After obtaining Query, Key, and Value in the multi-head attention block, the LIF function is adopted to generate spike-based features. Then the data is transmitted to feedforward blocks with linear layer to improve the information expression capability of T-SGAN. After the multi-head attention block and feedforward block, the Conv1D layer with (1, 1) kernel size is adopted to map the data with the shape of $(T, E)$ to the shape of $(T, C)$, in order to obtain the synthetic data with the same shape as the real data.

**Discriminator**. The spike-based discriminator is similar to the generator. The final layer of the discriminator networks employs the linear layer for decoding decision, which computes the firing rates of neurons through the whole temporal dimension to determine the results the classification.

**Model Training.** When training the T-SGAN, we use binary cross-entropy error as the loss function to update parameters. We set real_label as a list filled with number 1, representing input data of the discriminator are true while fake_label as 0 for synthetic data. We mark the input of the generator as $d$. Then $G(d)$ represents the output of the generator and $D(G(d))$ denotes the judgment of the discriminator. Then the loss function for the discriminator and generator are as follows:

$$\text{real\_loss} = \text{BCELoss}(D(\text{real\_data}), \text{real\_label}), \tag{4}$$

$$\text{fake\_loss} = \text{BCELoss}(D(G(d)), \text{fake\_label}), \tag{5}$$

$$\text{dLoss} = \text{real\_loss} + \text{fake\_loss}, \tag{6}$$

$$\text{gLoss} = \text{BCELoss}(D(G(d)), \text{real\_label}). \tag{7}$$

where real_loss represents the loss between output of discriminator and real label when the input is real data, while fake_loss is the loss when the input is fake data. We choose dLoss to update parameters of discriminator, to expect discriminator to perform correct judgment between real data and fake data. gLoss is used on the generator, which will cheat discriminator, judging the data generated by generator as real data.

Figure 2: (a)The spatial and temporal attention component used in the generator of T-SGAN. (b)Temporal dimension segmentation method. The temporal segmentation component is introduced to support the synthetic long but sparse VIP neuronal response generation.

During the training, random vectors are put into generator and generate fake data which has the same shape with real data. Then real data and fake data are feed into discriminator and firstly update parameters of discriminator using loss described above. And then it will repeat the operations to obtain fake data and finally using gLoss to update parameters of generator. That forms a complete updating in one epoch.

### 3.1.3 TEMPORAL DIMENSION SEGMENTATION

Though we have implemented a window function to the original data and reduced time points, it's still relatively long for the spike-based transformer to process. To address this, we design temporal dimension segmentation before every multi-head attention block. As depicted in Figure 2, considering a matrix $I \in R^{T \times E}$, we partition it into several segments with the shape (seg_num, points_num, E). Here, Seg_num represents the number of segments, and points_num denotes the number of time points within each segment. The product of these two values equals $T$. Each segment independently undergoes multi-head attention. Following segmentation, the number of time points decreases to points_num, significantly reducing the computational load compared to the original $T$.

However, a challenge arises as one time point can only compute relevance with others within the same segment. Consequently, two highly relevant points may be unable to establish contact if they are divided into different segments. To address this issue, we introduce a repartition step. The data is redivided into a distinct number of segments with the shape (seg_num', points_num', E). Here, Seg_num' and points_num' retain the same meanings as mentioned earlier but are not necessarily equal to their previous counterparts. Importantly, Seg_num and seg_num' should not be multiples of each other to prevent independent computation among neighboring time points. Due to the repartition, at least two multi-head attention blocks are now incorporated in both the generator and discriminator.

There exists work with similar partition mechanism, such as Swin Transformer (Liu et al., 2021). We have to emphasize the major difference between our work and Swin Transformer. Firstly, Swin Transformer is implemented in visual tasks, whose data usually form as 2-D images. Our temporal segmentation is applied on time-series data, usually 1-D sequences. Secondly, what's the most biggest difference is data processing. In Swin Transformer, through patch partition or patch merging, channels of images will increase while height and width decresae. But ours is different. We choose to add a dimension, for example, data with shape $(T, C)$ transfers to $(T1, T2, C)$. The first dimension $T1$ is processed by SNN and the second dimension $T2$ is processed by spiking transformer. That's mean that we not only select time points in a segment to conduct self-attention but use one segment as a unit to be processed by SNN, which truly combines SNN and transformer structure.

### 3.1.4 SPATIAL RELEVANCE EXTRACTION

In the multi-head attention block, after we execute temporal self-attention, we add a spatial self-attention module to extract spatial relevance. As the signals are sampled from cells in the same brain region, the spatial relationship between those cells needs to be considered to improve the effectiveness of the decoding process. While temporal self-attention computes relevance among time points, spatial attention computes relevance among different cells in a segment. In one multi-

head attention block, we describe its operation as:

$$X = X + \text{dropout}(\text{att}(\text{layernorm}(X))), \tag{8}$$
$$X = \text{transpose}(X), \tag{9}$$
$$X = \text{transpose}(X + \text{dropout}(\text{att}(\text{layernorm}(X)))), \tag{10}$$

where transpose means a transposition of temporal dimension and spatial dimension. The $att$ function is as the same as the normal self-attention which contains Query, Key and Value. We choose dropout layer to avoid over-fitting.

## 3.2 RECURRENT SNNs FOR HEADING DIRECTION DECODING

Due to the spatial and temporal dynamics of VIP neurons responding to the heading direction, our decoding process aims to design SNNs that adaptively identify the relatively important time periods within a long sequence and specific sensitive neuron populations among all VIP neurons corresponding to different directions, significantly influencing decoding results.

The architecture of the recurrent SNN decoders consists of input layer, hidden layer and output layer. Firstly, following the generation of synthetic data by the generator, we combine it with the original data and feed the merged dataset into the input layer of the SNN decoders for the training process. Subsequently, the hidden layer employs recurrent connection between different time steps. And the temporal attention and spatial attention are integrated into recurrent SNN classifiers, in order to respectively select crucial time points and cells crucial to the results. Further, prior to direction classification, a window function is applied to the data, similar to the one employed during T-SGAN training. The distinction lies in the window size used in classification is larger than that used in T-SGAN training. This adjustment is to ensure data integrity, as a small window size may compromise the completeness of the data. Therefore, we divide the sequence into several segments and seek the crucial time in units of one segment rather than individual time points, because judging temporal importance point by point is inefficient and unreliable for a long sequence. Then we sum up all numbers in a segment and pass them through a sigmoid activation. If the segment is deemed important, it will maintain its original numbers; otherwise, it will converge to be infinitely close to zero. The final decoding directions are achieved by population decoding over these segments.

## 4 EXPERIMENTS AND RESULTS

Our experiments are primarily conducted on two datasets. Dataset1 is derived from 90 cells in the brain of a single monkey, while Dataset2 is collected from 210 cells across the brains of three monkeys. All experimental data collecting procedures received ethical approval and adhered strictly to the guidelines outlined in the National Institutes of Health Guide for the Care and Use of Laboratory Animals.

**Experimental settings.** To ensure robust evaluation, we partition the entire dataset into a training set and a test set using a 3:1 ratio. The training set is dedicated to training and generating within the T-SGAN framework, while the test set exclusively serves for direction decoding testing. Subsequently, we combine the generated data with the training set, leading to a merged dataset that is further divided into training and validation sets. This division follows a 2:1 proportion and employs a three-fold cross-validation method. After undergoing 50 cycles of training, validation, and testing, we compute the average accuracy across these iterations, considering it as our final result. Additional details about datasets, along with comprehensive experimental settings, are provided in the Appendix. In our analysis, three types of typical classifiers with three layers, including fully connected SNN (FCSNN), LSTM, and Recurrent SNN (ReSNN) with spatial and temporal attention, are employed to show the generalization of the proposed decoding framework. Each decoding classifier incorporates a hidden layer, and we treat the number of hidden neurons as a hyperparameter, systematically observing its impact on overall accuracy.

Table 1: Overall accuracy comparison

| Models | Hidden size | dataset1 | | | | | dataset2 | | | | |
|--------|-------------|---------|------|------|------|------|---------|------|------|------|------|
| | | without | with | seg1 | seg2 | seg3 | without | with | seg1 | seg2 | seg3 |
| FCSNN | 64 | 87.70±0.45 | **92.30**±0.34 | 94.55±0.28 | 95.85±0.23 | 89.60±0.26 | 85.45±0.48 | **88.45**±0.40 | 96.40±0.27 | 97.45±0.29 | 93.10±0.30 |
| | 128 | 87.90±0.45 | **94.45**±0.37 | 96.20±0.32 | 95.55±0.28 | 91.60±0.36 | 86.50±0.51 | **91.40**±0.36 | 97.55±0.24 | 98.75±0.22 | 94.45±0.30 |
| | 256 | 88.45±0.49 | **94.75**±0.35 | 95.05±0.29 | 96.60±0.25 | 91.20±0.30 | 86.30±0.44 | **91.55**±0.50 | 97.60±0.24 | 98.70±0.23 | 95.00±0.30 |
| LSTM | 32 | 78.80±0.49 | **79.60**±0.46 | 94.05±0.26 | 93.25±0.27 | 82.80±0.33 | 68.25±0.55 | **70.20**±0.48 | 93.80±0.31 | 90.80±0.26 | 75.40±0.27 |
| | 64 | **85.30**±0.51 | 82.15±0.47 | 95.45±0.25 | 95.20±0.25 | 89.90±0.29 | 76.10±0.52 | **77.40**±0.51 | 98.10±0.29 | 93.95±0.33 | 82.15±0.32 |
| | 128 | **87.25**±0.51 | 86.10±0.47 | 94.95±0.29 | 95.70±0.28 | 91.80±0.31 | 79.40±0.49 | **81.25**±0.46 | 97.95±0.22 | 95.60±0.27 | 84.65±0.30 |
| ReSNN | 64 | 90.45±0.50 | **90.60**±0.39 | 93.95±0.27 | 94.55±0.28 | 86.65±0.30 | 81.75±0.54 | **88.75**±0.42 | 96.45±0.29 | 95.20±0.32 | 90.05±0.35 |
| | 128 | 93.35±0.53 | **94.05**±0.36 | 96.40±0.24 | 96.40±0.26 | 89.90±0.28 | 87.60±0.48 | **92.25**±0.39 | 98.25±0.24 | 98.85±0.22 | 93.65±0.28 |
| | 256 | 93.55±0.52 | **95.20**±0.29 | 95.95±0.28 | 97.35±0.23 | 91.45±0.29 | 91.00±0.46 | **93.15**±0.43 | 99.00±0.18 | 99.35±0.16 | 94.85±0.31 |

Table 2: Comparison with TTS-GAN

| Models | without | Methods | |
|--------|---------|---------|---------|
| | | T-GAN | TTS-GAN |
| GLM | 50 | **52.5** | - |
| DNN | 85 | **90** | - |
| FCSNN | 88.45 | **94.75** | 91.0 |
| LSTM | **87.25** | 86.1 | 84.5 |
| ReSNN | 93.55 | **95.2** | 93.75 |

Table 3: Accuracy on different segments

| Models | Hidden size | (5,4) | (50,40) | (50,40,16) | (5,4,2) | (20,16) | (20,16,10) |
|--------|-------------|-------|---------|------------|---------|---------|------------|
| FCSNN | 64 | **90.95** | 78.325 | 47.4 | 89.125 | 90.75 | 48.45 |
| | 128 | **93.0** | 81.325 | 48.75 | 91.85 | 89.95 | 50.275 |
| | 256 | **92.9** | 82.1 | 50.55 | 91.5 | 89.85 | 51.2 |
| LSTM | 32 | **78.15** | 76.65 | 43.875 | 77.95 | 76.8 | 44.55 |
| | 64 | 83.15 | 81.25 | 47.65 | **83.7** | 82.05 | 46.625 |
| | 128 | 84.55 | 81.5 | 47.8 | **84.85** | 83.75 | 47.65 |

**Overall performance comparison.** We evaluate the test accuracy by comparing the decoding performance with and without generated data on two datasets. The hidden sizes for FCSNN and ReSNN are set to 64, 128, and 256, while for LSTM, they are set to 32, 64, and 128 for fair comparison. During the classification process, we identify that a window size of 60 is optimal for FCSNN, whereas 300 is more effective for LSTM and ReSNN. Consequently, these configurations are adopted for our classifiers. As depicted in the left two columns of each dataset in Table 1, our T-SGAN demonstrates a significant accuracy improvement in heading direction decoding datasets. Focusing on results greater than 90% and excluding comparatively lower results, we observe a maximum accuracy boost of 6.3% in dataset1 and 5.25% in dataset2. This outcome affirms the exceptional performance of our T-SGAN in data generation and augmentation. Moreover, among the three classifiers, ReSNN outperforms others, achieving the highest accuracy of 95.2% in dataset1 and 93.15% in dataset2. These findings underscore the substantial benefits conferred by the integration of recurrent architecture and spiking neurons in decoding neural signals based on spikes. Furthermore, comparing the performance with and without generative data with LSTM, we observe that the combination of our T-SGAN and SNN decoder more closely aligns with selective segmentations' performance. This observation underscores the practical advantages of our T-SGAN and SNN decoder in decoding neural signals through data augmentation.

**Comparison with other GAN model.** To assess the effectiveness of the proposed T-SGAN, we conducted a comparative analysis with TTS-GAN on dataset1, one of the leading time-series data generating models, as presented in Table 2. Across most of classifiers including GLM (Generalized Linear Model), DNN (deep neual networks ), FCSNN, and ReSNN, the decoding model with synthetic data generated by T-SGAN consistently achieves higher accuracy compared to TTS-GAN using the same classifiers. However, in the case of LSTM, the accuracy without generated data surpasses both GANs. This discrepancy suggests a potential incompatibility of LSTM with spike data, which further emphasizes the significance of designing the SNN-based T-SGAN model to improve the decoding performance for spike-based neural data.

**The effectiveness of temporal dimension segmentation.** We conducted experiments on temporal dimension segmentation to assess the impact of segment partition, as discussed in Section 3.1.3. In the experiment, the window size for the classifier is set to 120. We represent the segment numbers using a tuple (a, b), such as (5, 4), indicating the sequence is initially divided into 5 segments and then redivided into 4 segments. As outlined in Table 3, the tuple (5, 4) yields the highest accuracy of 93%, and other tuples like (5, 4, 2) and (20, 16) also demonstrate commendable results. However, for larger tuples, the results are comparatively poorer. The reason lies in that as the segment numbers increase, the number of time points of each segment decreases proportionally. Consequently, the segment becomes too short to encompass all highly related time points, which makes it difficult to exploit the advantage of capturing long-range dependencies in sequences in T-SGAN.

Table 4: Accuracy of without/with spatial self-attention module

| Models | Hidden size | Window size(ms) | | | | | | | |
| --- | --- | --- | --- | --- | --- | --- | --- | --- | --- |
| | | 30 | | 60 | | 120 | | 300 | |
| FCSNN | 64 | 89.75 | **90.0** | 91.75 | **92.3** | 90.85 | **90.95** | 85.75 | **86.35** |
| | 128 | 91.25 | **91.3** | 94.125 | **94.45** | 92.65 | **93.0** | **86.1** | 85.6 |
| | 256 | **91.75** | 91.65 | 94.25 | **94.75** | 92.7 | **92.9** | **83.15** | 83.05 |
| LSTM | 32 | 70.15 | **70.4** | 76.4 | **76.5** | 77.625 | **78.15** | 79.5 | **79.6** |
| | 64 | 75.1 | **75.15** | 80.65 | **80.8** | **83.25** | 83.15 | **82.5** | 82.15 |
| | 128 | **72.65** | 72.5 | 82.325 | **82.55** | 83.925 | **84.55** | 85.95 | **86.1** |
| ReSNN | 64 | 87.25 | **87.5** | 90.1 | **90.15** | **92.0** | 91.9 | 90.2 | **90.6** |
| | 128 | 88.15 | **88.2** | 92.5 | **93.1** | 92.95 | **93.2** | 93.65 | **94.05** |
| | 256 | 90.75 | **91.05** | 93.6 | **93.7** | 94.3 | **94.7** | 94.85 | **95.2** |

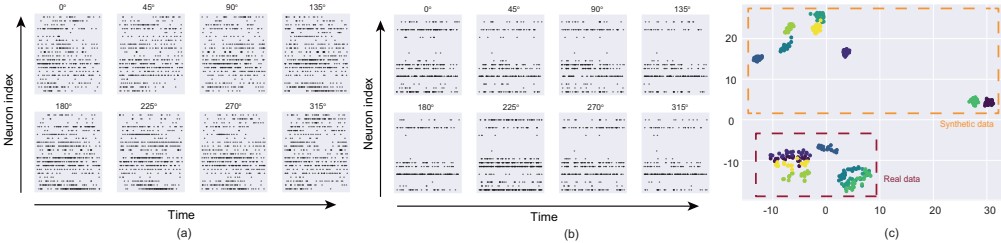

Figure 3: Visualization of real data and synthetic data. (a) Real data. (b) Synthetic data by T-SGAN. (c) The t-SNE analysis.

**The effectiveness of spatial self-attention.** We conducted a comparative analysis of the T-SGAN with and without the spatial self-attention module. As shown in Table 4, the left and right columns of each window size represent the decoding accuracy without and with the spatial self-attention module, respectively. With the spatial self-attention module, FCSNN, LSTM, and ReSNN achieved the best results of 94.75%, 86.1%, and 95.2%, respectively, compared to 94.25%, 85.95%, and 94.85% without this module. Meanwhile, each classifier exhibited improvement with the data augmentation by T-SGAN, demonstrating that the spatial self-attention module authentically enhances the performance of T-SGAN.

**Visualization of generated data.** Additionally, we assess the performance of T-SGAN through qualitative visualizations presented in Figure 3. The evaluation involves a comparison of spiking data among specific neuron cells between real and generated data. To highlight the features of synthetic data and its relationship with real data, we provide visualization examples in (c), where data point distributions are mapped to two dimensions using t-SNE. The visualizations demonstrate that synthetic data exhibits a more pronounced inter-class distance for different classes, contributing to the enhanced training of the SNN decoder model for heading direction classification. Meanwhile, the generated data brings diversity, which could enhance the model's generalization ability and facilitate the generation of novel instances that contribute to improved decoder training.

**Power consumption estimation.** Furthermore, as mentioned above, compared with DNNs, the proposed SNNs has the biologically plausible features of event-driven, which enables low power consumption and easy-implementation in neuromorphic hardware. As shown in appendix, the energy consumption estimation indicates that the SNNs could achieve higher decoding accuracy while consuming less energy than ANNs.

## 5 CONCLUSIONS

In this work, we build a Temporal Spiking generative adversarial networks (T-SGAN) that is able to generate time-series data that are suitable for spiking data. We implement temporal dimension segmentation and spatial self-attention to our T-SGAN to promote the quality of generated data. We also add spatial and temporal attention mechanism in the recurrent SNNs to raise the decoding accuracy. We conduct heading direction decoding from VIP neuronal responses to show that the proposed T-SGAN successfully generates synthetic data and promote decoding accuracy of recurrent SNNs.

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

## A   APPENDIX: SUPPLEMENTARY DATASETS DESCRIPTION AND EXPERIMENTAL SETTINGS

**Datasets.** In the sampling process, the monkeys get stimulated from eight horizontal directions and the response of VIP neurons are recorded. The collected dataset contains 160 trials in total, 20 trials for each direction. In each trial, it contains 1200 time points to reflect signal changes. The datasets have the shape of (160, 1200, 90) for dataset1 and (160,1200,210) for dataset2. The labels have the shape of (160, 1) to represent the eight directions with integers from 1 to 8.

**Experimental settings.** When training our T-SGAN, we set the window size mentioned in Section 3.1 as a constant number 30ms, so the input sequences' length is 40. We apply Adam to optimize

the parameters of the generator and discriminator and train the networks with 1000 epochs. Finally, in the whole mixed data, real data has the same proportion as synthetic data of the VIP population neuronal responses. In the heading direction decoding process, we choose several window sizes mentioned in Section 3.2. The unit of these windows is 'ms', as the sample frequency of our data is 1000HZ.

## B APPENDIX: PARAMETERS USED IN EXPERIMENTS

Table 5: Parameters description

| name | description | value |
|---|---|---|
| ge_bin_size | window size used in T-SGAN | 30 |
| epoch | epochs to train T-SGAN | 1000 |
| label_emb | embedding size of labels | 100 |
| latent_dim | size of random vector | 50 |
| hidden_size | hidden size of decoding models | 32/64/128/256 |
| bin_size | window size used in decoding models | 30/60/120/300 |

In our SGAN model, we use ge_bin_size to represent the window size in T-SGAN, which is 30 ms. We train the SGAN in 1000 epochs and the embedding size of label is 100. The size of random vector, which we use latent_dim to represent, is set to 50. And in the decoding models, we implement hidden size of 32, 64, 128 or 256 and window size of 30, 60, 120 or 300. We list the parameters used in our experiments in Tab. 5.

Then, the surrogate gradient of $g_{(x)}$ is approximated based on the derivative of spike activities of $g'(x) = \frac{\alpha}{2(1+(\frac{\pi}{2}\alpha x)^2)}$, to solve the non-differential problem of discrete spikes firing behavior, where $\alpha$ is set to be 2.0. Based on the above formulation and the gradient computing in (Wu et al., 2018), the update of synaptic weights can be obtained by the gradient descent rules.

## C APPENDIX: THE EFFECTIVENESS OF TEMPORAL AND SPATIAL ATTENTION

Table 6: Accuracy of without/with temporal and spatial attention

| Models | Hidden size | Window size(ms) | | | | | | | |
|---|---|---|---|---|---|---|---|---|---|
| | | 30 | | 60 | | 120 | | 300 | |
| FCSNN | 64 | **90.2** | 90.0 | 91.95 | **92.3** | **90.8** | 90.95 | **89.35** | 86.35 |
| | 128 | 91.1 | **91.3** | 93.95 | **94.45** | 92.5 | **93.0** | **89.95** | 85.6 |
| | 256 | 91.0 | **91.65** | 94.2 | **94.75** | 92.55 | **92.9** | **89.875** | 83.05 |
| ReSNN | 64 | 87.25 | **87.5** | **90.25** | 90.15 | 91.6 | **91.9** | 89.95 | **90.6** |
| | 128 | 88.15 | **88.2** | 92.85 | **93.1** | 93.0 | **93.2** | 93.75 | **94.05** |
| | 256 | 90.75 | **91.05** | 93.2 | **93.7** | 93.95 | **94.7** | 94.5 | **95.2** |

The results of Tab. 1 indicate that the results are significantly impacted by different segmentation of sequences. Therefore, it is crucial to incorporate an attention mechanism that can select the functional segment adaptively. As shown in Tab. 6, we compare accuracies of FCSNN and ReSNN about whether the classifiers contain temporal and spatial attention. The left and right column of each window size represents accuracy without and with attention mechanism, respectively. In FCSNN, the best result without attention mechanism is 94.2% and it promotes 0.55% for the result with attention mechanism to an accuracy 94.75%. In ReSNN, the best results of the two terms are 94.5% and 95.2%, which has a 0.7% boost.

# D  APPENDIX: THE ENERGY CONSUMPTION ESTIMATION

We conduct the energy estimation of the proposed SNNs decoding models on Linux server with the max CPU frequency of 5800MHZ. We compute power consumption of ReSNN and LSTM based on the method used in (Zhou et al., 2022) with the window size of 120 ms and hidden size of 128. The total energy of ReSNN and LSTM is $E_{AC} * 76800 + E_{MAC} * 2222080$ and $E_{MAC} * 2222080$ respectively, where $E_{AC}$ represents energy of one spike-based accumulate operation and Emac represents one multiply-and-accumulate operation. Time of ReSNN and LSTM to process a batch is 0.023s and 0.008. Thus, the power is $(E_{AC} * 76800 + E_{MAC} * 2222080)/0.023$ and $(E_{MAC} * 2222080)/0.008$ respectively. As described in (Zhou et al., 2022), $E_{AC}$, about 0.9pJ, is much lower than $E_{MAC}$ which is 4.6pJ. Hence, compared with ANNs, the SNNs could achieve higher decoding accuracy while consuming less energy.

# E  APPENDIX: THE PERFORMANCE OF DIFFERENT TEMPORAL SEGMENTATION

Given the apparent temporal dynamics in VIP population responses, evident from the computation of firing rate variation trends, there tends to be a specific temporal segment most responsive to heading direction within each sequence. To investigate the influence of different segments, we extract various segments from each sequence, as illustrated in the right three columns of Table 1. Specifically, seg1, seg2, and seg3 represent segments with time points of 1∼720, 240∼ 960, and 480∼1200, respectively. In both datasets, seg1 and seg2 exhibit high accuracy, while seg3 performs relatively poorly, highlighting the distinct temporal dynamics associated with different segments. However, extracting the most responsive neural signals is challenging due to different time-varying responses caused by different experimental paradigms. Comparing the results with and without generative data, we observe that the combination of our T-SGAN and SNN decoder more closely aligns with segmentation performance. This observation underscores the practical advantages of our T-SGAN and SNN decoder in decoding neural signals through data augmentation.

