

Figure 1. Example neural responses in VIP to the vestibular heading. (A) For an example VIP neuron, the spike raster of each trial (first row) and peristimulus time histograms (PSTHs, second row) are shown for each of the eight heading directions. Red and green vertical lines indicate onset and offset of the vestibular stimulus respectively. The blue curves indicate fits to the PSTHs. The density and distribution of neural spikes from the neuron changed with the vestibular stimulation over time in a trial relative to the baseline. (B) Average spike density functions (SDFs) across trials in each direction are shown for the neuron in (A), which preferred around 0∘ heading. Peak time of the spiking responses located around midcourse of the stimulation period, suggesting the neuron encoded velocity component of the vestibular heading. And the neuron showed different firing rates to the eight headings and preferred 0°. (C–E) Format is as in (B) but for another three VIP neurons with different preferred headings: around 90°, 180° and 270°, respectively. Some neurons may encode acceleration component of the vestibular heading (C), whereas some neurons may encode both velocity and acceleration components (E). Therefore, the responses of VIP neurons to the vestibular heading showed substantial spatial and temporal dynamics.

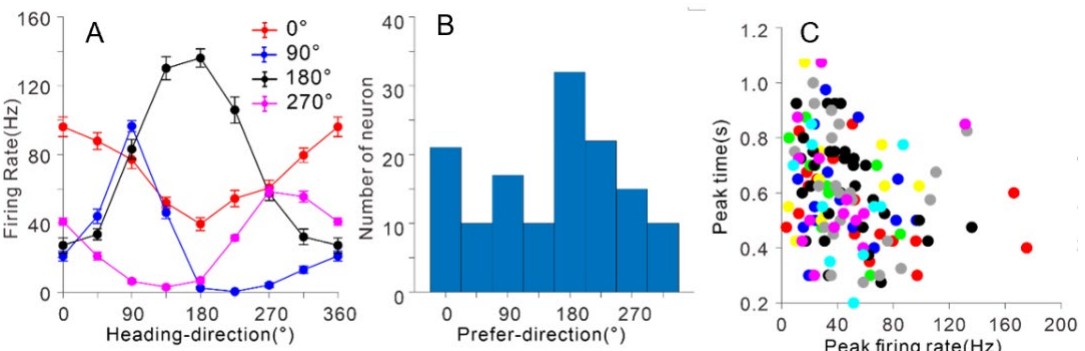

Figure 2. (A) Tuning curves are shown for the neurons with different preferred headings (around 0°, 90°, 180° and 270° respectively). For all VIP neurons (pooled across monkeys) with significant heading direction selectivity, (C) The peak firing rate and time are shown in a scatter plot. Each symbol indicates data from a VIP neuron. Color indicates preferred headings with color codes as in Fig. 4B. For all VIP neurons (pooled across monkeys) with significant heading direction selectivity distributions of the preferred headings (B), the peak firing rate (C) are shown.