# OpenReview forum: "Temporal Spiking Generative Adversarial Networks for Heading Direction Decoding"
_ICLR.cc/2024/Conference — Submitted to ICLR 2024_

### Official Review · Reviewer_1g5p · 2023-10-30

**Soundness:** 2 fair
**Presentation:** 2 fair
**Contribution:** 2 fair
**Rating:** 5
**Confidence:** 4

**Summary:**

The authors propose a spiking neural network that reproduces neural data from the VIP cortical area to compensate for the scarcity of this experimental data. In order to design this model, they use a sophisticated network based on spiking transformers, which they then evaluate by computing the decoding performance. They show a modest but significant improvement in performance.

**Strengths:**

The paper is relatively clear. The results are interesting. My first major comment is that the paper proposes an application to neuroscience using neural networks to improve decoding capabilities. This is a completely original and promising line of research that is not often presented at this conference and should be encouraged.

**Weaknesses:**

I find that the neuroscience aspect is very lightly touched upon. For example, the authors do not show a selectivity curve for the neurons studied, but only the decoder performance results in the form of tables of numbers. The impact of the paper could be improved if the authors presented in a synthetic way the principle of encoding the direction of the head in the VIP area, as well as the different results as improved by their methods. Finally, one of the interesting results of this paper is to show that different decoding architectures give different performances. This result could shed light on the inherent complexity of the neural representation underlying the distributed representation of head orientation in the population of neurons studied, but I lack this analysis.

**Questions:**

I think the paper could be improved by addressing the following points:
- " With the sparse spike-form Query, Key, and Value, its computation becomes more efficient.": please justify
- in the model, why use firing rates when you claim that the representation is spiking?
- Figure 1 is too small, "sythetic" > "synthetic"
- Figure 3: the description of the t-SNE analysis is vague and the results seem rather negative, this is hardly discussed in "Visualisation of generated data". Please clarify.

Minor:
- The syntax of the paper did not allow me to fully follow the arguments. I have not taken this into account in my evaluation, but the authors should use a service, even an automatic one, that allows clarification of certain points.
- The LaTeX formatting of the paper could be improved. In particular, quotations in the text should be enclosed in parentheses, e.g. using ``citep''. Text appearing in equations ("real", "data", ...) should be formatted as text, e.g. using `\text'.

---

> ### Author Response · Authors · 2023-11-22
>
> Thanks for your efforts on our paper. We have polished the manuscript as suggested.
>
> **Q1**" Sparse spike-form Q, K, V is more efficient." **A:** Thank you for bringing up this point. In the traditional Transformer model, the computation involves dense matrices for Query, Key, and Value, wherein each element is a continuous float number, contributing to the dense computation. In contrast, the spike-based Transformer introduces a novel approach by utilizing a sparse spike form for the self-attention computation. This sparsity is strategically harnessed during the computation of Query, Key, and Value matrices, leading to a reduction in the total number of Multiply-and-Accumulate (MAC) calculations required during the attention computation [1, 2]. This optimization not only ensures a more efficient computation process but also proves particularly advantageous for spike-based models. As shown in the section D of APPENDIX in the revision about the energy consumtion estimation, the spike-based model consumes less energy than the dense none-spike computation. Therefore, exploiting the inherent sparsity of neural events allows us to enhance computational efficiency without compromising the overall performance of the model.
>
> [1] Yao, M., et al. Spike-driven Transformer. In Thirty-seventh Conference on NeurIPS 2023.
> [2]Zhou, Z., et al. Spikformer: When Spiking Neural Network Meets Transformer. ICLR 2023.
>
> **Q2:**"Why claim spiking representation?" **A:** Thank you for your insightful question. The use of firing rates in our model, despite claiming a spiking representation, is a design choice driven by practical considerations. The real and generated spike firing data exhibit sparsity in the temporal dimension, making processing the data at each time point inefficient. By employing a relatively small time binning, the binned data also presents the spike form with 0 or 1 in each bin, as well as enhancing data generation efficiency. The firing rate representation, which reduces the length of the neuronal data, is also particularly beneficial for the Transformer-based structure, as demonstrated in Table 3. This approach addresses challenges associated with handling long-term dependencies and reduces computational complexity. The temporal dimension segmentation in our paper further supports these design considerations.
>
> **Q3:**"Figure 1"  **A:** The figure 1 has been modified to be clear and correct in the revision.
>
> **Q4:**"The description of the t-SNE" **A:** Thanks for your question. Here the similarity reflects in the responses of some key neurons with high response intensity. The assessment of the generated synthetic data quality extends beyond mere similarity to the original data. Diversity in the generated data is a crucial factor, enhancing the model's generalization ability and facilitating the generation of novel instances that contribute to improved decoder training. In our paper, the visualization of generated data is discussed with a comprehensive consideration of both similarity and diversity, and the subsequent performance improvements in decoders are presented and analyzed (refer to Table 1 and 2).
>
> **Q5**"Syntax and latex writting."  **A:** Thanks for your suggetion. We polished the whole manuscript and marked some key modified content red in the revision.
>
> **W1:** "The analysis of the principle of encoding the direction of the head in the VIP area. " **A:** Thanks you for pointing that. We supplement the example neuronal data analysis in the  figure of the supplementary PDF fileThe responses of VIP neurons to heading stimulation indeed show high heterogeneity. Different VIP neurons show single or double peaks in a trial, have different response latencies and are selective to different heading directions and sensory modalities (vestibular or visual).  Except for the peaks in a trial, other time periods of neuronal signals during this trial may not be relevant to the heading direction decoding (the irrelevant information could cause noise).
>
> **W2:** "Inherent complexity of the neural representation."  **A:** Thank you for your valuable suggestion. The revised Figure 3 illustrates that the generated synthetic data exhibit distinct neural representations of head orientation when compared to the real collected data. Specifically, the figure demonstrates that the generated neural data predominantly captures the neuronal responses of specific neurons, while other neurons remain relatively inactive throughout the entire time window. This observation underscores the varying spatial dynamics within the population of neurons. Additionally, the generated neuronal responses, as indicated by spike firing frequency, manifest different intensities across various time points when contrasted with the real data. This discrepancy highlights the differences in temporal dynamics between the generated and real data.

---

### Official Review · Reviewer_w27Z · 2023-10-30

**Soundness:** 4 excellent
**Presentation:** 4 excellent
**Contribution:** 3 good
**Rating:** 8
**Confidence:** 4

**Summary:**

The paper introduces a unified spike-based decoding framework for studying the neuronal responses in the ventral intraparietal area (VIP) related to heading directions. Given the limited biological data for VIP neuronal responses, the paper presents a novel approach using spiking neural networks (SNNs) and generative adversarial networks (GANs) to create synthetic neuronal response data. This synthetic data is then employed to improve the decoding of heading directions, with a focus on temporal and spatial dynamics. The proposed temporal spiking generative adversarial networks (T-SGAN) generate realistic synthetic data, enhancing the decoding accuracy of recurrent SNNs. The paper demonstrates that this framework leverages the benefits of biological plausibility and low energy consumption of SNNs in neuronal response decoding.

**Strengths:**

1. The paper introduces an innovative approach to studying neuronal responses in the VIP, combining spiking neural networks and generative adversarial networks. The use of SNNs for decoding and T-SGAN for data augmentation is a unique and original contribution to the field.
2. The paper presents a well-structured and detailed methodology, making it easy for the reader to understand the proposed framework. The experiments conducted on biological datasets from monkeys provide robust evidence of the framework's effectiveness.
3. This research addresses a critical challenge in neuroscience, offering a new approach for decoding neuronal responses in the VIP. The potential applications of this framework, including its low energy consumption advantages, make it significant in the field.

**Weaknesses:**

1.  It would be beneficial to discuss the limitations and potential biases associated with using synthetic data for training neural networks. Additionally, addressing the potential discrepancies between synthetic and real data would provide a more comprehensive understanding of the framework's applicability.
2.  The paper mentions that SNNs were trained to decode heading directions using the generated synthetic data. More information about the SNN training would enhance the reproducibility of the research.

**Questions:**

see weaknesses

---

> ### Author Response · Authors · 2023-11-22
>
> Thank you for your thoughtful comments and suggestions, and thank you for engaging with us to make our work stronger.
>
> **W1** "The limitations and potential biases associated with using synthetic data for training neural networks." **A:** Thank you for raising an important point. We acknowledge the limitations and potential biases associated with using synthetic data for training neural networks. While synthetic data, generated by models like T-SGAN, can enhance the diversity of training sets and contribute to improved generalization, it is essential to recognize that the synthetic data may not fully capture the complexities and nuances present in real-world datasets. Addressing potential discrepancies between synthetic and real data is crucial for a comprehensive understanding of the framework's applicability. We plan to conduct further investigations and analyses, comparing model performance on synthetic and real data, to provide a more nuanced discussion on the generalization capabilities and limitations of our proposed approach. Additionally, we will explicitly detail these considerations in the revised manuscript to ensure transparency and thorough evaluation of the framework.
>
> **W2**"More information about the SNN training would enhance the reproducibility of the research."  **A:** Thank you for your valuable suggestion. To enhance the reproducibility of our research, we have provided additional information about the SNN training process, including details on surrogate gradients. This information is now available in Section B of the revised manuscript's appendix. We believe these additions will contribute to a clearer understanding of the SNN training methodology, supporting the reproducibility of our proposed approach.

---

### Official Review · Reviewer_rvuq · 2023-10-30

**Soundness:** 1 poor
**Presentation:** 1 poor
**Contribution:** 1 poor
**Rating:** 1
**Confidence:** 4

**Summary:**

The paper proposes a spiking neural network (SNN) approach to decode heading direction from neural activity in parietal cortex of monkeys. In addition, it proposes a GAN-based data augmentation strategy. The authors fit their model on two datasets of 90 and 210 neurons from area VIP in monkey parietal cortex and show that it outperforms two alternative (SNN-based?) baselines.

Unfortunately the paper is written so poorly that I found it impossible to figure out what exactly is happening and why. I think this paper needs a full rewrite and resubmission at a future venue.

**Strengths:**

I have a hard time listing any, since I simply did not understand the paper.

**Weaknesses:**

1. Extremely poor writing; paper is basically impossible to follow
 1. Motivation for SNN-based approach unclear
 1. Motivation for GAN-based data augmentation unclear
 1. Simple baselines (e.g. linear model) missing

**Questions:**

While I think I understood the overall objective (decode heading direction), I had a really hard time figuring out why they take the SNN as opposed to standard neural networks or even simpler methods such as (generalized) linear decoding. Despite best efforts and multiple reads of abstract, introduction etc. I could not follow the reasoning.

In terms of evaluation, the most straightforward baselines such as a (generalized) linear model are missing. Thus, it is not clear what any of the numbers in the paper mean.

The effect of the GAN is questionable, as the differences reported in Table 1 are small and there are no error bars reported.

---

> ### Author Response · Authors · 2023-11-22
>
> Thank you for engaging with us to make our work stronger. We address individual questions and concerns below. We would greatly appreciate it if you took the new experiments and clarifications below into account during discussions.
>
> **W1.1 and Q1:** "Motivation for SNN-based approach" **A:** The motivation behind employing Spiking Neural Networks (SNNs) in the proposed T-SGAN and recurrent SNN approach for neural spike data decoding lies in the unique characteristics of SNNs that align with the nature of spike-based neural signals. SNNs inherently capture temporal dynamics more effectively due to their event-driven nature, making them well-suited for processing spike sequences. This is crucial for decoding neural signals, which often involve intricate temporal patterns. Furthermore, SNNs are known for their energy-efficient computation, making them suitable for applications where power consumption is a concern, such as in neuromorphic hardware. This aligns with the goal of developing efficient models for processing neural spike data.
>
> **W1.1 and Q1:** "Motivation for GAN-based data augmentation." **A:** We appreciate the reviewer's inquiry regarding the motivation for integrating GAN-based data augmentation into our spike-data neural decoding framework. The rationale for employing GANs in this context is rooted in their unique capacity to address specific challenges associated with spike data. GANs are particularly suited for spike-data augmentation due to their ability to capture and reproduce intricate temporal patterns inherent in neural spike sequences. Spike data exhibits complex dynamics that are pivotal for decoding tasks, and GANs excel in capturing these intricate temporal dependencies. By leveraging the generative capabilities of GANs, our approach aims to enhance the diversity of synthetic spike sequences, ensuring that the generated data aligns with the authentic temporal intricacies observed in real neural recordings. Moreover, GANs are instrumental in mitigating the issue of data scarcity common in neural spike datasets. The generation of synthetic spike sequences allows us to substantially augment our dataset, providing the model with a more extensive set of examples to learn from. This is particularly advantageous in scenarios where obtaining large-scale labeled spike datasets is challenging. Additionally, the adversarial training paradigm of GANs enables the generation of realistic and contextually relevant spike patterns, contributing to a more effective and generalizable decoding model. The synthesized spike data not only serves as an augmentation strategy but also enriches the feature space, promoting the learning of discriminative features crucial for accurate decoding. In summary, the motivation behind incorporating GAN-based data augmentation lies in their ability to faithfully capture complex temporal patterns, address data scarcity issues, and provide a diverse and realistic synthetic dataset. These characteristics collectively contribute to the effectiveness and robustness of our spike-data neural decoding framework.
>
> **Q3**"In terms of evaluation, the most straightforward baselines such as a (generalized) linear model are missing. "  **A:** Thank you for your valuable suggestion. We have supplemented a comprehensive evaluation by including baselines such as generalized linear models (GLM) and artificial neural networks in the comparison. The updated results, now presented in the following table (and Table 2 of the revised manuscript), demonstrate that our proposed ReSNN model outperforms the linear model (GLM) in terms of decoding accuracy. We believe this addition strengthens the evaluation and provides a more thorough understanding of the performance of our proposed model.
>
>
> | Models | Without synthetic data | With synthetic data from T-SGAN |
> |--------|-------------|----------|
> | GLM    | 50| **52.5** |
> | DNN    | 85 | **90** |
> | FCSNN  | 88.45 | **94.75** |
> | LSTM   | **87.25** | 86.1 |
> | ReSNN  | 93.55  | **95.2**  |
>
> **Q4** "The effect of the GAN is questionable, as the differences reported in Table 1 are small and there are no error bars reported. " **A:** Thank you for your feedback. We have included error bars in Table 1 to provide a more comprehensive representation of the results. Regarding the effectiveness of T-SGAN, we conducted a  comparative analysis with TTS-GAN, a prominent time-series data generation model, under the settings with various classifiers, including GLM, deep neural networks, FCSNN, and ReSNN. Across most of these classifiers, the decoding model using synthetic data generated by T-SGAN consistently outperforms TTS-GAN in terms of accuracy. Notably, in the case of LSTM, the accuracy without generated data surpasses both GANs. This discrepancy suggests a potential incompatibility of LSTM with spike data, which further emphasizes the significance of designing the SNN-based T-SGAN model to improve the decoding performance for spike-based neural data.

---

### Official Review · Reviewer_2sPx · 2023-11-05

**Soundness:** 3 good
**Presentation:** 2 fair
**Contribution:** 3 good
**Rating:** 5
**Confidence:** 4

**Summary:**

The work proposes methods for augmentation of spike recordings and decoding them into heading directions for VIP brain area. In particular, the paper employs a spike transformer for generation of synthetic data of neuronal spiking time-series of VIP neurons. The synthetic data along with real data is then processed by spiking recurrent neural network (SNN) to decode the responses into  heading directions. The decoding with the proposed approach is compared against decoding without the approach on two proprietry datasets of recordings from monkeys VIP.

**Strengths:**

1. The approach proposes a decoding pipeline for VIP neural recordings based on spiking time-series and spiking neural networks without transformation to and from firing rates.

2. The proposed approach appears to enhance the eventual decoding accuracy.

3. The approach proposes temporal dimension segmentation through self-attention as a component of Spikeformer to deal with long sparse time-series of spikes.

**Weaknesses:**

1. Contribution of the augmentation (synthetical generation) vs. SNN decoding to increase in accuracy is unclear.

2. Related to previous point, the presented augmentation approach in principle appears to be plausible for other neural recordings, i.e. other brain areas and tasks. However, the presented work does not consider generalization and is too specialized towards VIP and decoding heading direction.

3. Data visualization shows differences in synthetically generated spike trains data vs. real data. Also t-SNE embedding shows very different embedded points. These differences are not explained. On the contrary, spike trains are claimed to be similar.

4. There is no full description of experimental setup of data acquisition, how data was preprocessed, validated and whether this will part of this work.

5. While it is possible to understand the content, the manuscript is not well-written and includes grammatically incorrect sentences and typos.

**Questions:**

Would appreciate authors answers to W1-W5 listed above. In particular:

Re. W1 &W2, how one would discern the contribution of the augmentation vs. the special decoder?
Does the approach have independent merit as an augmentation method?
If real data would replace synthetic data would accuracy increase be similar?

Re. W3, it is unclear why the authors claim that the generated spike trains are similar to real spike trains. How do authors assess similarity.

---

> ### Author Response · Authors · 2023-11-22
>
> Thank you for your thoughtful comments and suggestions, and thank you for engaging with us to make our work stronger.
>
> **Q1 && W1**"How one would discern the contribution of the augmentation vs. the special decoder" **A:**  Thanks for your question. Firstly, the contribution of the augmentation with the proposed T-SGAN is evaluated in Table 2, by comparing the decoding performance of T-SGAN with TTS-GAN (one of the leading time-series data generating models) under the same decoders of FCSNN, LSTM, and ReSNN. The experimental results show that the decoding model with T-SGAN achives superior accuracy compared to TTS-GAN, affirming the effectiveness of the augmentation contribution. Secondly, we assess the contribution of the ReSNN decoder by comparing its decoding accuracy with other decoders, including fully-connected SNNs (FCSNN), linear models, and LSTMs, under the same data augmentation model. As shown in both Table 1 and Table 2, the proposed ReSNN decoder outperforms other decoders, providing evidence for the effectiveness of the proposed decoding architecture.
>
> **Q1**“Does the approach have independent merit as an augmentation method”**A:** Yes. Firstly, the proposed T-SGAN offers independent merit as an augmentation method. It leverages spike-based computation, enhancing efficiency by reducing Multiply-and-Accumulate (MAC) calculations through sparse spike-based operations. This characteristic not only contributes to computational efficiency but also aligns with the suitability for implementation in neuromorphic hardware, as demonstrated in Section D of the Appendix in the revised manuscript. Secondly, T-SGAN is specifically designed for spike-based neural data augmentation. All computations in the augmentation model are spike-based, eliminating the need for data transformation. This spike-based approach has the potential to enhance performance, making T-SGAN a valuable and efficient augmentation method for spike-based neural data.
>
> **Q1**"If real data would replace synthetic data would accuracy increase be similar?"**A:** Yes, I think so. The question about replacing synthetic data generated by T-SGAN with real data and observing a similar increase in accuracy is an insightful one. In our experiments, we focused on the effectiveness of the proposed T-SGAN for data augmentation, and its performance was evaluated in conjunction with real data. However, it's crucial to note that the classification accuracy tends to benefit from a larger dataset. The increase in accuracy with a larger dataset is a common trend observed in machine learning models, including those leveraging synthetic data for augmentation. Therefore, while real data augmentation could lead to accuracy improvement, the magnitude of the improvement is likely to be influenced by the specific characteristics of the dataset and the capacity of the underlying model to generalize from the increased volume of real data. We appreciate this perspective and will further investigate and discuss this aspect in our revised manuscript.
>
>
> **Q2 && W3**"Data augmentation." **A:** Thanks for pointing out this. The contribution of the data augmentation could be measured from similarity and diversity [1]. Here the similarity reflects in the responses of some key neurons with high response intensity (figure 3 (a) and (b)). The assessment of the generated synthetic data quality extends beyond mere similarity to the original data. Diversity in the generated data is a crucial factor, enhancing the model's generalization ability and facilitating the generation of novel instances that contribute to improved decoder training. In our paper, the visualization of generated data is discussed with a comprehensive consideration of both similarity and diversity, and the subsequent performance improvements in decoders are presented and analyzed (refer to Table 1 and 2).
>
> [1] Lucic, M., et al. Are GANs Created Equal? A Large-Scale Study. In Advances in Neural Information Processing Systems (NeurIPS), 2018.
>
> **W4**"The description of experimental setup of data acquisition, how data was preprocessed, validated and whether this will part of this work." **A:** We would explain more details about the data acquisition, however, the data collection is not the part of this work. In detail, three adult rhesus monkeys weighing 6-9kg were used. Surgically, MRI compatible T-bolts were implanted in the head to fix a headcap, through this headcap to immobilize the monkey relative to a customized chair during experiments. Scleral coils were implanted in eyes for tracking eye positions as described previously. The recording sites in VIP were precisely targeted by lower electrodes through predetermined holes in a recording grid fixed on top of the skull of monkeys. Only neurons with clear response modulations to the vestibular stimulus were further tested with the protocol described above. Data were collected from 210 neurons in six hemispheres of the three monkeys.

---

### Meta-Review · Area_Chair_TViq · 2023-12-09

**Metareview:**

This paper proposes a novel framework for decoding heading direction from neural activity in the VIP area of the monkey parietal cortex. It uses a combination of spiking neural networks (SNNs) and generative adversarial networks (GANs) to overcome the limitations of limited biological data. The synthetic data generated by the T-SGAN enhances the decoding accuracy of recurrent SNNs, demonstrating the effectiveness of the proposed approach. This framework offers significant advantages over traditional methods, including biological plausibility and low energy consumption, making it a promising tool for studying the neuronal basis of heading direction perception.

Despite its good points, the paper suffers from several major weaknesses. First, it fails to clarify the contribution of its synthetic data generation and SNN decoding to the improved accuracy. Second, it lacks motivation for both its SNN-based approach and GAN-based data augmentation, and it does not include simple baselines for comparison. Third, it does not discuss the limitations of using synthetic data, and it omits crucial details about the SNN training process. Fourth, it lacks a strong neuroscience connection, neglecting to present neuron selectivity curves and perform in-depth analysis of the results. Additionally, it misses the opportunity to analyze the performances of different decoding architectures and shed light on the neural representation of head direction. These crucial flaws render the paper unsuitable for publication without significant improvements.

**Justification For Why Not Higher Score:**

Despite its good points, the paper suffers from several major weaknesses. First, it fails to clarify the contribution of its synthetic data generation and SNN decoding to the improved accuracy. Second, it lacks motivation for both its SNN-based approach and GAN-based data augmentation, and it does not include simple baselines for comparison. Third, it does not discuss the limitations of using synthetic data, and it omits crucial details about the SNN training process. Fourth, it lacks a strong neuroscience connection, neglecting to present neuron selectivity curves and perform in-depth analysis of the results. Additionally, it misses the opportunity to analyze the performances of different decoding architectures and shed light on the neural representation of head direction. These crucial flaws render the paper unsuitable for publication without significant improvements.

**Justification For Why Not Lower Score:**

N/A

---

### Decision · Program_Chairs · 2024-01-16

Reject